# A Systematic Review on the Link between Animal Welfare and Antimicrobial Use in Captive Animals

**DOI:** 10.3390/ani12081025

**Published:** 2022-04-14

**Authors:** Maria Rodrigues da Costa, Alessia Diana

**Affiliations:** 1Epidemiology Research Unit, Department of Veterinary and Animal Science, Northern Faculty, Scotland’s Rural College (SRUC), An Lòchran, 10 Inverness Campus, Inverness IV2 5NA, UK; maria.costa@sruc.ac.uk; 2Department of Comparative Pathobiology, Purdue University, 625 Harrison Street, West Lafayette, IN 47907, USA

**Keywords:** antibiotic stewardship, companion, farm, laboratory, zoo, pigs, cattle

## Abstract

**Simple Summary:**

The threat of antimicrobial resistance is a global health concern, and the misuse of medications is often considered a major contributor. Thus, judicious antimicrobial stewardship in captive animal species (i.e., farm, zoo, companion, and laboratory animals) is paramount and should rely on effective strategies for the reduction of antimicrobial use (AMU). Despite the relationship between welfare, health and productivity, the role that animal welfare can play in such a reduction has been poorly investigated, especially with regards to empirical evidence. This systematic review aimed to summarise the available body of research on the link between animal welfare and AMU in captive species. The low number of publications retrieved from the search, with 76% of them published in the last five years, revealed the knowledge gap pertaining this topic. The majority of publications was on farm animals, suggesting a relevance of the topic for this group, with most of the work (82%) looking at the effect of animal welfare on AMU, rather than the opposite. Overall, better animal welfare was found to be associated with lower AMU. Studies were mainly carried out in EU, likely due to its well-known role as being the avant-garde of animal welfare and AMU. Further research is needed to support these findings, especially concerning other captive species beyond farm animals.

**Abstract:**

This systematic review aimed to assess the link between animal welfare and antimicrobial use (AMU) in captive species (i.e., farm, zoo, companion, and laboratory animals) and its effect. Studies empirically examining the effect of welfare on AMU or vice versa were included. Studies in wild animals were excluded. A total of 6610 studies were retrieved from PubMed^®^ and Web of Science^®^ in April 2021. Despite finding several papers superficially invoking the link between welfare and AMU, most did not delve into the characteristics of this link, leading to a small number of publications retained (n = 17). The majority (76%) of the publications were published from 2017–2021. Sixteen were on farm animals, and one publication was on laboratory animals. Most of the studies (82%) looked at the effect of animal welfare on AMU. The body of research retained suggests that, in farm animals, better animal welfare often leads to lower AMU, as was hypothesised, and that, generally, poor welfare is associated with higher AMU. Additionally, AMU restrictions in organic systems may prevent animals from receiving treatment when necessary. Limitations of this study include focusing only on empirical research and excluding non-peer reviewed evidence. More research is needed to corroborate these findings, especially on the link between animal welfare and AMU in other captive species.

## 1. Introduction

Antimicrobial resistance (AMR) is a global human and animal health threat [1]. Inappropriate or excessive use of antimicrobials (AMU) may result in the development of resistance to these substances, and to the subsequent inefficacy of the treatments administered to tackle infectious diseases. In humans, AMR is causing over 33,000 deaths every year just in the EU [2], and it is already a leading cause of death worldwide [3]. Thus, although antimicrobials are fundamental for the health of humans and animals, their misuse poses a paramount risk to the development of resistant bacteria. This link has been widely confirmed by the European Medicines Agency (EMA) and the European Food Safety Authority (EFSA) in their joint report [4].

The use of medications in veterinary medicine together with human medicine [4], are major contributors to the development of AMR. The role played by the veterinary sector has been mainly reported in studies on farm animals [5] which, among all the categories of animals raised and/or managed by humans, are likely to be the highest consumers of antimicrobials [6]. However, evidence of resistant bacteria has been described in all captive species (i.e., companion, laboratory (lab), and zoo animals) [7,8,9] making them reservoirs of AMR. For instance, an investigation by Álvarez-Pérez et al. [10] reported that zoo species such as chimpanzees and Iberian ibex carried strains of *Clostridioides difficile* exhibiting resistance to antimicrobials commonly used in both veterinary and human medicine. The study of Ishihara et al. [11] also identified an association of AMU with the spread of resistant *Escherichia coli* among zoo animals. One of the roles of modern zoos is to promote wildlife conservation through breeding and reintroduction programs [12]. However, these practices may become a potential route of dissemination of resistant bacteria not only among zoos worldwide but also into the wild. Indeed, reintroduction of zoo species to their natural environment can contribute to the spread of AMR to the wildlife [13]. A similar scenario was also observed in studies on companion and lab animals such as that of Loncaric and colleagues [14] where different companion animals (e.g., dogs, cats, rabbits) receiving antibiotic therapy had higher chance to develop resistant bacteria. The authors also observed that ‘hospitalized animals’ had higher risk to carry methicillin-resistant *Staphylococcus* sp. Another example is the study of Yamanaka et al. [15] who found laboratory mice showing resistance to several classes of antibiotics such as macrolides and fluoroquinolones.

In the companion, laboratory, and zoo animal groups, AMU seems low when compared to farm animals [16]. For instance, Joosten et al. [17], who investigated AMU and AMR in companion animals across three EU countries, reported that 81% of the animals included in their study did not receive any antimicrobial treatment. Thus, in these species the issue to address regarding AMU is not a matter of ‘quantity’ but of ‘quality’, since the most common medications used were critically important antimicrobials [17]. Despite this, a more prudent antimicrobial stewardship in all group species is needed and should rely on the development of effective strategies that can help to address an inappropriate AMU [18]. Greater knowledge on AMU and on potential risk factors for its use in all captive animal groups is then pivotal to achieve this goal. 

At the same time, animal welfare is nowadays an ethical and societal demand. Indeed, providing appropriate welfare standards is considered a priority for animals living in captivity including those in the agriculture sector, in zoo institutions, those used in research, and pet animals. A set of rules are in place, both in EU and internationally [19,20], for the protection of all animal categories from farm [21], to companion [22], to laboratory [23], and to zoo species [24]. These rules/legal frameworks establish the minimum welfare standards to be respected. 

The relationship between animal health, animal welfare, and productivity is well acknowledged and scientifically recognized [25], as stated by the OIE in its Guiding Principles for Animal Welfare, which declared ‘*a critical relationship between animal health and animal welfare*’ and emphasised that ‘*improvements in animal welfare can often improve productivity and food safety*’ [26]. Yet, animal health can still be perceived as separated from animal welfare, with the latter being considered more as a cost than a benefit to exploit. Instead, they depend on each other and can be considered as ‘two faces of the same coin’, thus making the concept of ‘One Welfare’ the natural extension of the ‘One Health’ approach [27,28]. Both concepts recognise the interconnection between humans, animals, environment, and conservation to support a more global sustainable development [27]. Integrating these two approaches in research studies allows for a more holistic perspective towards certain areas of interest. In particular, it will permit to gather more evidence on direct and indirect benefits of incorporating the field of animal welfare to other disciplines to untangle the AMR threat [27,28]. 

Nevertheless, despite such promising benefits, the role that animal welfare can play in the reduction of AMU has been poorly investigated especially with regards to empirical evidence. In their recent report, the Food and Agriculture Organization of the United Nations (FAO) stated that improved health and welfare would make animals less prone to contract infectious diseases, thus minimising the need for antimicrobials [29]. The necessity for more research on this argument is then evident. However, despite several publications that widely emphasised and theoretically discussed the importance of exploring such a relationship in animals kept in captivity [29,30,31,32], the extent of scientific work where this link has been demonstrated and/or studied in the literature is unclear. The importance of involving multiple disciplines when investigating this topic seems also to be a returning argument of discussion, with it (i.e., multi-disciplinarity) having been identified as a key tool to better understand such a relationship while also providing additional information on AMU among captive species. Deeper knowledge on the link between animal welfare and AMU, will greatly contribute to the development of effective strategies for a more judicious AMU in veterinary medicine. 

In this systematic review, we aimed to gather research that explores the link between animal welfare and AMU in captive species. In particular, we focused on those studies that investigated either the impact of improved/poor animal welfare on AMU or vice versa—i.e., that of reduced/increased AMU on welfare indicators. This work is paramount to synthetize the empirical knowledge available so far, to compare the state-of-the-art between different captive animal groups, and to generate valuable information to target gaps in the literature regarding the aforesaid topic.

The research question framed was “Does animal welfare have an impact on antimicrobial use, or vice versa, in captive species (farm, zoo, companion and laboratory animals)?”. The population targeted were all captive animals within those four groups and the outcomes expected were empirical evidence of the effect of animal welfare on antimicrobial use or vice versa.

## 2. Materials and Methods

This systematic review aimed to explore the link between animal welfare and antimicrobial use in captive animals. This work was framed in the context of a special issue entitled ‘A Multidisciplinary Approach to Unveil the Link between Animal Welfare and Antimicrobial Use in Captive Animals’ for the journal Animals. The methods employed were based on the PRISMA (Preferred Reporting Items for Systematic Reviews and Meta-Analyses) statement [33]. PRISMA’s checklist for systematic reviews is included as Appendix A.

Captive animals were grouped into: Farm animals, Zoo animals, Companion animals, and Laboratory (Lab) animals. The search string built for the searches was composed of three segments. Each segment assembled relevant keywords and synonyms. An ‘OR’ operator was used within segments whereas an ‘AND’ operator was used between segments. The structure of the search strings follows example 1:*Search string* = *(animal welfare) AND (antimicrobial use) AND (animal group)*


The animal welfare and antimicrobial use segments (keywords used of the search) were common to all groups. The last segment referred to the animal group (i.e., farm animals, zoo animals, etc.) and to a list of the most representative animals included in that group. Two online databases, PubMed^®^ and Web of Science^®^, were selected to conduct the literature searches. All searches were conducted in April 2021. The searches were restricted to the title and abstract and included only peer-reviewed studies in English. No time limit was imposed. The detailed search strings employed in each database are available in the Appendix A. 

The quality of the search was assessed by checking whether previously (manually) identified papers of interest (sentinel papers) were retained in the systematic searches. The search results in both databases were imported using EndNote^®^. The same reference manager was used to remove duplicates. Table 1 presents the inclusion and exclusion criteria defined to screen titles and abstracts and for the full text evaluation.

An initial sampling of 80 records (20 records in each animal group) for training purposes was performed. These records’ titles and abstracts were screened by the two assessors (the two authors) in parallel to practice the application of the inclusion and exclusion criteria. After this step, the authors discussed the results and refined the established criteria. Additionally, 20 records (5 in each group) were assessed in conjunction in real time to further validate the criteria. After this session, the assessors conducted the title/abstract evaluation of all records blindingly and independently. Any disagreements were discussed. Once the paper selection for full text analysis was finalised, the papers selected were retrieved and the two co-authors (in parallel and blinded to each other’s decisions) read the full texts using the same eligibility criteria (Table 1). Exclusion of records had to be agreed by both authors. 

Study quality was ensured through the methods applied. First, this systematic review targeted only peer-reviewed publications, narrowing down the body of research to include only scientifically sound articles previously assessed by peers. Second, the article screening steps were devised to minimise bias of selection and ensure assessors were in agreement during different stages of the process. Risk of bias was not formally assessed in studies included in this review. Since the results were qualitative, a narrative description supported by graphs and tables was the preferred method of synthesis. Therefore, the risk of bias was addressed on a group basis, anchored in individual examples, and reported in the discussion.

The data within the final records included in this work was extracted onto a database (stored in a Microsoft Office Excel^®^ spreadsheet). Data extraction accounted for the year of publication, the journal name, and the topic (the latter information was retrieved from ‘Scimago JR’ by selecting the first subject area of the journal), the country where the study was carried out, the country of the institution where the studied was developed (regarding first author), and the animal group studied. Other information, such as welfare indicators used, the route of antimicrobial administration, the direction of the study (i.e., whether it tested the impact of welfare on AMU or vice versa), and whether there was an effect, was also included. Graphical, tabular, and narrative commentary were the methods of synthesis used in the results’ section.

## 3. Results

### 3.1. Systematic Literature Search

The systematic search retrieved a total of 9448 publications of which 5808 publications were from the database ‘Web of Science’ and 3648 publications were from the database ‘PubMed’. Table 2 shows the list of records included in all stages of the systematic review process.

The screening process and subsequent agreement of the authors led to the final retention of 17 publications for this systematic review. Of those, 16 were publications on farm animals and only one on lab animals [34]. No publications were retained for companion and zoo animals. The list of the papers excluded after full text evaluation and the reason for their exclusion are available in the Appendix A. In total, five sentinel papers were used, and all figured in the searches. Finally, while evaluating publications, the authors had an agreement rate of 99.2% and of 100% in the title/abstract evaluation phase and on the full text evaluation phase, respectively.

### 3.2. Characteristics of the Publications

The majority of publications (76%) were published in the last five years (2017–2021) with 52.9% of them published in the last two years only. The earliest publication was published in 2001. Denmark was the most represented country in terms of study location (29.4%) and of the first author’s main affiliation (23.5%) followed by Italy (17.5% for both categories). Only one publication [35] reported a study carried out in more than one country. Figure 1 shows the distribution of publications across the years (A) and the countries of the studies and of the first author (B).

In total, 12 journals were used to publish the publications gathered in this systematic review. ‘Animals’ was the leading peer-reviewed journal where these works were published (23.5%), followed by ‘Journal of Dairy Science’ and ‘Acta Veterinaria Scandinavica’ in *ex aequo*, each accounting for 11.7% of the publications. Hence, these three journals published almost half (46.9%) of the publications gathered. 

Journals having as main topic of research ‘Agriculture and Biological Sciences’ were the most selected for these publications (41.2%), followed by journals whose main topic was ‘Veterinary Medicine’ (29.4%). Table 3 and Table 4 list the journals where the selected papers were published and the related topics, respectively.

A total of 16 out of 17 publications investigated the link between animal welfare and antimicrobial use in farm animals. Specifically, pigs were the most studied farm animals (41.2%) followed by dairy cattle (29.4%). Table 5 shows the number of publications across the different animal groups and the species studied in the 17 records selected.

### 3.3. Type of Welfare Indicators Assessed and Route of Administration of Antimicrobials

The majority of publications (82.3%) used animal-based measures (e.g., somatic cell counts, behavioural observations, mortality) to assess the welfare status of the animals while parameters related to the housing and management conditions such as stocking density, bedding material and animal handling were used as welfare indicators in 58.8% (10 out of 17) of the publications. Both categories were used as indicators of animal welfare in seven publications but in the group of farm animals only. The use of parenteral administration of antimicrobials was the most common route reported in almost 60% of the publications while (41%, 7 out of 17) publications reported in-feed or water administration as the main route. Only one publication reported the use of a local/topic AM as a route of administration [36]. The type of indicators used to assess the welfare of captive animals and the route of administration of antimicrobials are presented in Table 6.

### 3.4. Direction and Effect of the Study

The majority of publications (82.3%) investigated the impact of animal welfare on antimicrobial use while only three of them [37,38,39], all from the farm group, were focused on the impact of reduced and/or increased use of antimicrobials on welfare indicators in pigs, dairy and beef cattle, respectively. A total of three out of 17 publications, which were all investigating the effect of welfare on AMU, did not report any effect [40,41,42]. Results are presented in Table 7.

#### 3.4.1. Publications Debating the Effect of Animal Welfare on AMU

Out of the 11 publications studying the impact of animal welfare on AMU which reported an effect, five were in pigs, five in cattle (beef and dairy) and one was in laboratory animals—monkeys. Kaiser et al. [36] studied the effect of using rubber mats and daily zinc ointment application to treat shoulder ulcers in 304 sows from three herds, compared to the daily spray of chlortetracycline (CTC). Sows were examined 1 day after farrowing and paired according to shoulder ulcer score. Results showed a significantly smaller size of the shoulder ulcer on days 14 and 21 after farrowing on sows treated with the rubber mat and the zinc ointment when compared to those which were treated with CTC spray. Isomura et al. [43] studied the effect of biosecurity and animal welfare indicators on AMU in feed across 38 Japanese pig farms. Space allowance in finishers, pre- and post-weaning mortality risk were used as proxies for animal welfare. Results showed that post-weaning mortality risk was positively associated with the use of in-feed doxycycline and amphenicols, but not with total AMU. A similar epidemiological study was conducted in Finland to analyse the effect of biosecurity and welfare standards on AMU in finisher pig farms [44]. The study used data from 2011 to 2013, which was extracted from a national database (Sikava), which gathers data on AMU, meat inspection, animal welfare and farm characteristics from more than 95% of pig production in Finland. The analysis showed that farms with average or poor drinking equipment, deficient enrichment and poor pen conditions combined with a low space allowance had an increased number of AM treatments per pig. The authors concluded that there is strong evidence to suggest that “*by improving biosecurity and welfare at pig farms, antimicrobial use can be reduced*”. In Italy, Tarakdjian et al. [45] analysed AMU data from 36 pig farms and, retrospectively, related it with their husbandry practices. The authors reported that farms engaged in welfare-friendly pig production systems were characterised by a 38% lower AMU as compared to conventional farms. Finally, Nielsen et al. [46] compared AMU records from 2016 to 2018 from three different pig production systems (organic, conventional free-range, and conventional indoor) in Denmark. Here, too, the authors found that welfare-label production systems had a significantly lower AMU when compared to conventional systems. Additionally, more than 30% of the welfare-label farms enrolled in the study did not use antimicrobial treatments, when compared to only 16% of the conventional indoor pig farms. All of the publications on pigs mostly referred/related to AMU in-feed, with exception for Kaiser et al. [36].

Two publications reported a positive effect of welfare-friendly labels on the AMU of fattening calves in Switzerland [47] and in Italy [48]. The former study found that the Swiss “outdoor veal calf” concept had an AMU five times lower when compared to control farms and further investigated whether this reduction was associated or not with “decreased animal welfare, i.e., that sick animals were not left untreated”. Results showed that calf health was effectively improved in the new “outdoor veal calf” system, having less respiratory and gastrointestinal diseases. In the latter study, Diana et al. [48] analysed the effect of welfare standards and biosecurity practices on AMU and also reported that lower AMU (reduction in treatment frequency) was observed with improved level of welfare standards in beef cattle (calves for fattening). Three publications studied the effect of welfare on AMU in dairy cattle. Vaarst et al. [49] compared AMU and key figures for production and udder health between organic and conventional Danish dairy herds and found that the duration of AM treatment of acute mastitis was shorter (on average) in the organic herds compared to the conventional herds. The authors reported that there was no difference in the incidence of mastitis treatments or somatic cell counts between the two types of herds. However, “*the calculated mean bulk milk somatic cell count level was higher (…) in the organic herds, and more cows had acute or chronic elevated SCC*”. In 2006, Vaarst et al. [50] interviewed 12 organic dairy producers and compared the frequency of mastitis and bulk tank somatic cell counts between farms phasing out AMU and farms with limited use. Mastitis treatments were less frequent, and the somatic cell count was smaller in the former group of herds compared to the latter. The authors stated that farms in the former group had a long-term plan to tackle antimicrobial use, which included higher welfare standards, despite all farms in this study being organic producers. Finally, Ivemeyer et al. [35] described the impact of an animal health and welfare planning (AHWP) protocol on AMU on 128 European dairy farms (from six EU countries, and the UK) from 2008 to 2010. The implementation of AHWP reduced AMU treatment incidence and improved the udder health situation across all farms. All cattle publications referred/related to injectable AMU. Finally, one recent publication studied the effect of modified head caps that protect sutured skin margins after cranial implant surgery in macaques [34]. The results showed that, across two UK primate facilities, the use of the protective head cap promoted wound healing, and none required re-suturing, contrary to control cases in which that was necessary up to 30% of the monkeys. More importantly, monkeys wearing the head cap had reduced AM treatment (length) and analgesia.

Out of the three publications that studied the effect of animal welfare on AMU and reported no effects, one was in dairy cattle, another in pigs, and another in poultry. Firth et al. [40] compared defined course doses (dcdvet) for blanket and selective antimicrobial dry cow therapy on conventional and organic farms in Austria. The study reported that the difference between AMU on conventional and organic farms for dry cow therapy as a whole was not statistically significant. The only difference found was across farmers which used blanket approaches vs. those using selective treatments: those using selective treatments used fewer AMU. Wadepohl et al. [41] evaluated whether a herd health and welfare index—which was an assessment tool developed for a European project on AMR and transmission (EFFORT)—was related to AMU. Results showed that this index was not related to AMU in the field. Finally, Tarakdjian et al. [42] revealed that farms rearing broilers with a space allowance of 33 and 39 kg/m^2^ did not exhibit differences in terms of AMU. No differences across the two groups were found regarding mortality and feed conversion ratio.

#### 3.4.2. Publications Debating the Effect of AMU on Animal Welfare

Three publications studied the effect of AMU on welfare and reported an effect. Diana et al. [37] assessed the welfare of pigs reared with and without antibiotics in-feed on a commercial pig farm in Ireland and concluded that treated pigs were more likely to have tail lesions but less likely to have ear lesions than control pigs. No difference in health indicators and mortality rate was observed between the two groups, whereas the number of ear biting and aggressive behaviour was higher in treated pigs than in control pigs. Turner et al. [38] studied the effect of removing critically important antimicrobials, alongside with strategies to encourage and improve management and husbandry from UK dairy herds on cattle health and welfare. Production parameters, fertility, udder health and mobility data and culling rates were maintained and even improved after the removal of HP-CIA and there was also a reduction in the overall AMU in the seven farms in the study. In beef cattle, Bokma et al. [39] analysed 76 production cycles from 26 Belgian veal farms (2014 to 2016) and concluded that an increase in mortality when reducing AMU in veal calves could not be shown. This would suggest that reducing AMU did not affect calves’ welfare, expressed as mortality risk.

## 4. Discussion

### 4.1. The Link between Animal Welfare and Antimicrobial Use: A Poorly Studied Research Topic

This systematic review aimed to gather and explore manuscripts, reviews and case studies that empirically addressed the link between animal welfare and AMU in captive species. Only 17 publications studying the effect of welfare on AMU or vice versa were relevant for this review, with 16 out of those referring to the group of farm animals. The low number of publications retrieved with this search is revealing of the knowledge gap pertaining the link between animal welfare and AMU. Thus, research that considers such a relationship as fundamental for the reduction of antimicrobials [29] and the associated threat of AMR [27,28] is still at an early stage. On the other side, the disproportion of papers found in favour of farm animals is symptomatic of the relevance of this relationship for this group of captive species, likely due to the greater consumption of antimicrobials administered on-farm when compared to zoo, lab, or companion animals [16]. Indeed, animals raised under intensive farming conditions are highly susceptible to infectious diseases [51,52], which in turn may lead to greater AMU, also prophylactically. This aspect, together with a greater consumer awareness and demand towards antimicrobial-free products, [53,54] is of great motivation for scientists, farm stockholders and for the industry to advance knowledge on this research area.

The fact that no publications were retrieved for zoo and companion animals within the criteria defined, highlights the need for further research on the link between welfare and AMU in these groups of captive animals. Another possible explanation to justify an imbalance in the number of publications among these group species is that zoo, companion and laboratory animals are more likely to be treated with antimicrobials when their health and welfare is actually impaired by disease, while farm animals can also be treated at prophylactic level or for growth promotion, albeit the former is increasingly more restricted and the latter is limited to some countries such as India and Latin America [55]. Thus, for these groups, a more judicious AMU should be focused on strategies aiming at reducing the use of ‘highest priority critically important antibiotics’ (HP-CIA) and at the same time minimising AMU. Indeed, Joosten et al. [17] in their study reported that the most common classes of antimicrobials used in companion animals were those classified as CIA. Moreover, the need for research on the link between welfare and AMU is also essential because the accumulation and development of AMR in zoo, pets, and laboratory animals is a reality and it has been documented by several studies [8,10,15], making this an aspect that cannot be underestimated. In fact, the spread of AMR from captive animals to humans and/or the wildlife is facilitated by several factors such as the release of threatened species in the wild through international conservation programs carried out by the zoos [13] or, in the case of companion animals, due to the human–animal relationship between owners and pets [56]. Therefore, given the contribution of AMU to the increase in AMR [4], strategies for the reduction of CIA use are much needed also with regard to these group species and not just for farm animals. However, although no studies were found through our systematic review that described the benefits of improved animal welfare on the reduction of AMU in zoo and companion species, those related to farm (see Section 3.4) and lab animals [34] can lead us to assume that welfare improvements may be beneficial to prevent AMU needs also in these captive groups. Our assumption can also rely on the acknowledged impact that environmental enrichment, improved environment, and management practices can have on the incidence of disease and on the behaviour of these captive species. For instance, speaking of companion animals, cats without access to the outdoors seem to have increased incidence of (feline) lower urinary tract disease (FLUTD) [57]. Whereas, looking at the zoo animals, greater variety and frequency of environmental enrichments had positive effects on the elephants’ reproductive health, leading to improved ovarian cyclicity and prolactin level [58,59].

The papers selected for this systematic review were published over the past 20 years (i.e., 2001–2021), though most of the studies retained (13 out of 17 studies, 76%) were published in the last 5 years. This outcome emphasises how, despite an increase in publications, an empirical investigation of the link between animal welfare and AMU in captive animals is still at a preliminary stage. As a result, the potential of incorporating the discipline of animal welfare for a more holistic view concerning the development of strategies to reduce AMU in animals kept in captivity [28] needs further exploration.

### 4.2. Where Is This Research Being Conducted, What Has Been Investigated and Where Is It Published?

The location and first author affiliation of all but three publications included in this review were based in the EU. The three exceptions are one publication that presented a study carried out in Japan [43] and two from the UK [34,38]. This finding is somehow expected given the high standards of animal welfare required by European policy for all groups of captive species [60], to which the UK also complied with until 2020 (Brexit), and also due to the stricter AMU regulations applied in the EU compared to the rest of the world. Last but not least, the new Veterinary Medicinal Products Regulation (Regulation (EU) 2019/6), officially applicable since January 2022, bans the preventive use of AM and ensures that treatments are prescribed and administered only when there is a real need for the animals [61]. The EU is indeed recognised as a global leader on these topics and its advocacy has positively influenced other non-EU countries to follow its route [60]. Specifically, Denmark is among the leading countries in the EU for standards of animal welfare in captive animals, for instance it was among the first countries in the world to recognize—with a law that came into force in 2021—that all animals are sentient beings [62]. This, together with the strict rules on AMU for farmed animals, likely justifies why a third of the total publications had this EU country as the main location of the study. Indeed, given the high standards of animal welfare to be maintained, the Danes may be more prone to care about advancing research on this topic.

All journals selected for publication provide an open-access service which is an option of great importance because it allows for an easier dissemination of such poorly investigated topic. However, a lack of diversity on the topic of research of the journals selected was evident. Indeed, most publications (n = 14) were published in journals with similar topics and in particular on the areas of ‘agriculture’ and ‘veterinary sciences’, probably due to the fact that 16 out of 17 publications were on farm animals. Based on this finding, we argue that researchers should also consider discussing the link between animal welfare and AMU under the scope of other disciplines, such as food policy, sustainability, sociology, and bioethics to fulfil the ‘One Health, One Welfare’ approach regarding the role of interdisciplinarity and interconnection in research [27,28]. This would allow an inflow of other perspectives, positive and negative arguments for improving animal welfare to reduce AMU or for administering AM to treat sick animals more judiciously.

Among the farm animals, pigs were the most studied species followed by cattle. This result is partly expected because pig production is known for its large use of AM (i.e., pigs consumed 32% of the overall veterinary antimicrobials sold in Europe in 2018 [63]), but also because the AMU patterns in pig production, such as group treatments using in-feed medication, justify the need to reduce and/or to use AM more wisely. On the other hand, most cattle publications retained focused on dairy herds and principally on dry cow treatments which is a practice still commonly applied to whole herds [64].

### 4.3. Is the Link between Animal Welfare and AMU Contributing to the Reduction of Antimicrobial Use?

Although our overall discussion is mainly based on work made on farm animals, we can claim that most of the studies included in this review linked better animal welfare to reduced AMU. However, one of the most striking features of some of the studies selected was the use of welfare measures that are not sensitive enough to detect poor welfare beyond that one considered as life-threatening (i.e., mortality rates). That is the case of the studies by Isomura et al. [43], Bokma et al. [39] and Tarakdjian et al. [45]. Though these studies did detect a link between animal welfare and AMU—where welfare-friendly systems or better welfare conditions on the farm was associated with lower AMU—it is possible that these investigations were detecting the effect of very poor welfare on AMU rather than the effect of good welfare or substantial welfare improvements on the reduction in AMU. It is also important to notice that the results of each study are not always easy to extrapolate for wider contexts, with exception for observational studies including multiple farms, and that the results of each study are representative of the conditions of that particular study/farm. An example is the work by Tarakdjian et al. [42] where the effect of space allowance on broilers AMU was studied but no results were found. In this case, only one type of space allowance was compared to the control (33 vs. 39 kg/m^2^, with the latter being the minimum space allowance required by EU law). Care is needed when stating that improved welfare conditions were not associated with AMU. Therefore, aspects such as the design of the study, poor choice of welfare indicators and a broader/inclusive definition of the concept of animal welfare need to be taken into account when evaluating study results.

Another example of the evidence of the link between welfare and AMU was the positive results obtained in studies such as Kaiser et al. [36] and Perry et al. [34]. In this case, improving welfare by reducing positional discomfort/friction prevented wounds, and therefore, helped to avoid AMU. This is somehow considered as common sense, since poor welfare conditions may lead to open wounds, which in turn may open doors for infectious agents that can then only be treated with AMU. However, poor welfare-resulting wounds can be caused not only by friction (e.g., shoulder sores in sows) but also by the performance of damaging behaviours such as ear and tail biting in pigs. The latter are considered as indicative of stress and occur when animals are unable to perform their natural behaviour and cannot cope with their environment, or when they are living in a situation of chronic stress [65,66]. These behaviours are multifactorial in nature and barren/highly stocked environments, group composition, diet and regrouping strategies, and climate conditions are just some of the potential risk factors [67,68,69]. Therefore, many welfare issues seem to be related to the physical and social environment of the animals. This highlights how appropriate management practices can, in certain circumstances, be more beneficial in ensuring animal welfare than the administration of prophylactic antibiotics. As reported by Diana et al. [37], factors such as stocking density, pig weight and room temperature played a more significant role on the occurrence of some of the lesions related to pig welfare than in-feed medications. However, this was not the case of ear lesions. In-feed antimicrobials seemed to play a role in mitigating the severity of these welfare-related lesions, thus suggesting that AMU when used wisely can be essential, especially under intensive farming conditions.

Finally, organic and biologic systems, which are presumed to be welfare-friendly, had lower AMU when compared to conventional systems in many of the studies retained [46,47,49]. This was an expected result. However, the research gathered in this work also discusses the possibility that some organic or biologic farms may have lower AMU because they can be more reluctant to treat animals when necessary, fearing the loss of the organic/biologic label. This second hypothesis was actively checked in the study by Moser et al. [47], and results showed that animals were not being left untreated and calf health was effectively improved in the more welfare friendly system. This finding supports the idea that when tools such as higher welfare standards, lower stocking densities and the avoidance of painful procedures such as tail docking are efficiently implemented as part of the routine farm management, a reduction of AMU can be achieved without compromising the health of the animals. On the contrary, Vaarst et al. [49] found that the duration of AM treatment of acute mastitis was shorter in organic herds compared to the conventional herds but there was no difference in the incidence of mastitis treatments or somatic cell counts (SCC) between the two types of herds, and ‘*more cows had acute or chronic elevated SCC*’ in the organic herds. According to the authors, this suggests that the shorter duration of mastitis treatment with antimicrobials in organic farms ‘*may be explained by restrictions on use of antibiotics because the organic farmer are not allowed to make follow up treatments with antibiotics in the organic herd*’. Therefore, it is reasonable to also assume that a consistent restriction of AMU may eventually correspond to an impairment of welfare. Thus, further empirical research on organic systems is very much needed to clarify this aspect, as similarly stated by the EMA and EFSA in their joint report [70], especially under the light of the new veterinary prescription rules (Regulation (EU) 2019/6) which restrict preventative AMU in the EU [61]. Since other farming realities may end up following similar strict AMU rules as those followed by organic farms (i.e., administration of AM only when really needed), a greater understanding of the implications of improved animal welfare standards and management strategies on AMU is of extreme importance.

### 4.4. Limitations of the Study

We are aware of potential limitations of this study such as those encompassing the search methodology. In fact, the results of this systematic review are valid within the context of the inclusion and exclusion criteria as defined a priori. This means that papers which did not empirically refer to the link between animal welfare and AMU were rejected. This decision was made to focus on studies with an effective analysis of this link—with a study design prepared to assess the impact of welfare on AMU or vice versa—as opposed to debate the theoretical link and its implications. In addition, it is worth to mention that publications not included in PubMed^®^ and Web of Science^®^ as well as those published in journals that were not peer-reviewed, were not identified in our search. We agreed that this was the most suitable approach for this review because the search on international citation databases (e.g., Web of Science, PubMed and similar), it is the most common method used by the scientific community worldwide [71,72]. This way, we made sure that all publications included had already gone through peer-review and were fully scrutinised on scientific basis. Understanding and summarising the possible scenarios in which animal welfare interacts and impacts on AMU and vice versa, is a task still quite far to be fully elucidated, and deserves greater attention from the research community.

## 5. Conclusions

Despite several papers superficially invoking the link between animal welfare and AMU were originally retrieved (n = 6610), most of them did not investigate the topic empirically nor delved into the characteristics of the link, leading to a small number of final publications retained in this systematic review (n = 17). We conclude that evidence for this link remains scarce in the literature. As hypothesised, this work suggests that better animal welfare often leads to lower AMU, and this was especially the case reported for farm animals. Accordingly, some studies demonstrated that poor animal welfare was associated with higher AMU. However, judicious AMU may be necessary and inclusively lead to better welfare (i.e., having a protective effect) when animals are reared under intensive or conventional settings (i.e., minimum/legal welfare standards met). At the same time, AMU restrictions in organic farm systems may prevent animals from receiving treatments, when necessary, likely posing an extra risk of affecting their welfare. Therefore, more research is needed to corroborate these findings, especially with regards to the link between animal welfare and AMU in other captive species (i.e., zoo, companion, and laboratory), going beyond farm animals.

## Figures and Tables

**Figure 1 animals-12-01025-f001:**
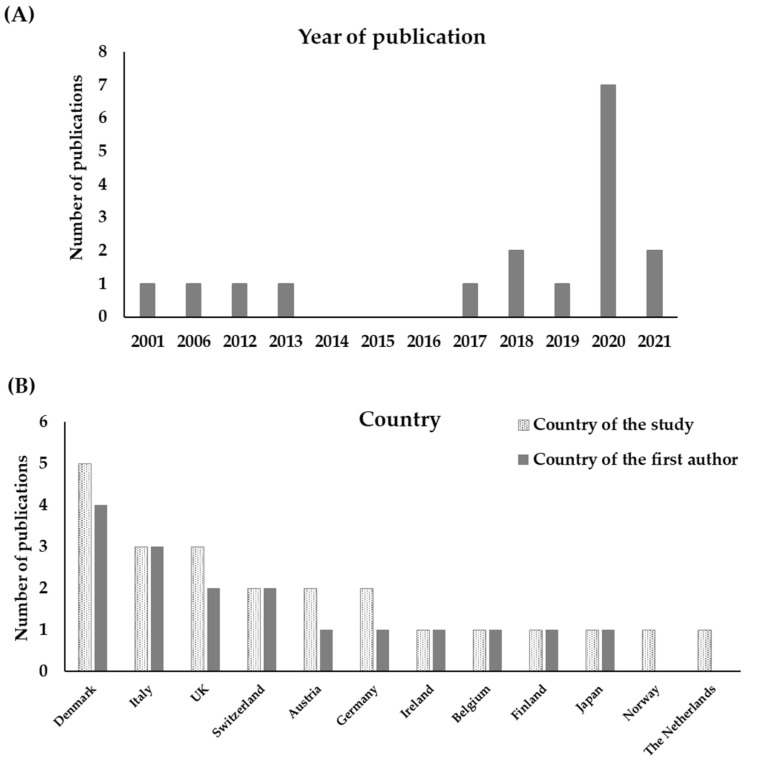
Number of publications by year of publication (**A**), country of the location of the study and first author affiliation (**B**). One publication had more than one country as location of the study.

**Table 1 animals-12-01025-t001:** Eligibility (inclusion and exclusion) criteria used for the screening of title/abstracts and full texts.

PICO ^1^	Inclusion Criteria	Exclusion Criteria
**Population**	Animal species being evaluated: must include (but not limited to) <target population> Unit of study (animal, batch, house, farm) Others: collections of farms, zoos, labs or national data referring to these captive species including companion animals, as long as the outcomes are debated in relation to each other	Papers studying wild animals and non-captive species
**Outcomes**	4. Focuses on the link/relationship between AMU and animal welfare with an empirical outcome	2. Papers not investigating empirically the link between AMU and animal welfare
**Others**	5. Language: English 6. Peer-reviews	3. Other languages 4. Other literature

^1^ PICO (participants, interventions, comparisons, and outcome(s))—framework to formulate research questions, following the methods proposed in the PRISMA statement [33].

**Table 2 animals-12-01025-t002:** List of records included in all stages of the systematic review for farm animals, zoo animals, companion animals, and lab animals.

Population	Total Records Retrieved	After Duplicates’ Removal	After Title and Abstract Screening	After Full Text Screening
Farm animals	5503	3669	25	16
Zoo animals	961	944	0	0
Companion animals	1390	893	0	0
Lab animals	1594	1104	1	1
Total	**9448**	**6610**	**26**	**17**

**Table 3 animals-12-01025-t003:** Number and percentage of publications by name of the peer-reviewed journal.

Journal *	n	%
Animals	4	23.5
JDS	2	11.7
Acta Vet Scand	2	11.7
Livestock Science	1	5.9
PLoS ONE	1	5.9
Veterinary Record	1	5.9
Animal	1	5.9
Scientific Reports	1	5.9
PVM	1	5.9
JVMS	1	5.9
TPAGGN	1	5.9
J Neurosci Methods	1	5.9

* JDS = Journal of Dairy Science; Act Vet Scand = Acta Veterinaria Scandinavica; PVM = Preventive Veterinary Medicine; JVMS = Journal of Veterinary Medical Science; TPAGGN = Tierärztliche Praxis Ausgabe G: Grosstiere—Nutztiere; J Neurosci Methods = Journal of Neuroscience Methods.

**Table 4 animals-12-01025-t004:** Number and percentage of publications by topic of research of the peer-reviewed journal.

Topic of the Journal *	n	%
Agriculture and Biological Sciences	7	41.2
Veterinary	5	29.4
Agriculture/ Veterinary	2	11.8
Multidisciplinary	2	11.8
Neuroscience	1	5.8

* Information on the topic of research was retrieved from ‘Scimago JR’ (https://www.scimagojr.com/, accessed on 31 August 2021) by selecting the first subject area of the journal.

**Table 5 animals-12-01025-t005:** Number and percentage of publications by group and species of animals studied.

Group Studied	Species Studied	n	%
	Pigs	7	41.2
Farm animals	Dairy cattle	5	29.4
(n = 16)	Beef cattle	3	17.6
	Poultry	1	5.9
Lab animals	Primate	1	5.9
(n = 1)			

**Table 6 animals-12-01025-t006:** Number and percentage of publications according to the type of welfare indicator assessed during the study and the route of administration of antimicrobials (AM). Publications can have more than one route of administration and welfare indicator investigated.

	Item	n	%
**Welfare indicator**	Animal based ^1^	14	82.3
	Housing and management ^2^	10	58.8
	Both indicators	7	41.2
**Route of**	Injections	10	58.8
**administration of AM**	In-feed/water	7	41.2
	Local AM	1	5.9

^1^ Indicators of animal welfare based on behavioural or physiological parameters (e.g., damaging behaviours, mortality, body condition score, somatic cell counts etc.);.^2^ Indicators of animal welfare based on the management practices and housing conditions (e.g., level of stocking density, presence of bedding material, feed and water availability etc.).

**Table 7 animals-12-01025-t007:** Number and percentage of publications by direction and effect of the study.

Direction of the Study ^1^	Effect of the Study ^2^		Total, n (%)
	Yes	No	
Animal welfare on AMU	11	3	14 (82.3%)
AMU on animal welfare	3	0	3 (17.7%)
**Total, n (%)**	**14 (82.3%)**	**3 (17.7%)**	**17**

^1^ The studies investigated the impact of animal welfare on antimicrobial use (AMU) or vice versa; ^2^ The studies investigated whether the impact of one of the two items (i.e., animal welfare and AMU) had an effect on the other one regardless the statistical significance.

## Data Availability

The raw data generated, analysed, and presented in this study are available upon request to the authors.

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
