# Peer review of "A Systematic Review on the Link between Animal Welfare and Antimicrobial Use in Captive Animals"

_animals, 2022, doi:10.3390/ani12081025_

Round 1

Reviewer 1 Report

This is an important and much needed review. It is very well constructed and written. There are some minor comments/suggestions for improvement.

General comments

Worth mentioning the new restrictions on preventative antibiotic use in the EU?

In my view, the potential for restricted treatments in organically farmed animals is worth mentioning but over-emphasised… There is a risk of withholding antibiotics in any farmed animal due to withdrawal periods for milk and meat for example.  I do agree that certain labelling, like raised without antibiotics have potential to lead to withholding of antibiotics when needed, but most organic labels don’t prohibit use, they would follow the same withdrawal periods as those used in conventional farming. This is why it’s worth mentioning the new restrictions in the EU, since all farming could end up following similarly strict AMU rules to that of organic farming to tackle the issue of AMR, and this could also lead to limited use when needed making welfare improvement very important.

Line by line comments

L11-12 – AMU is there but not spelled out. To be clear, suggest adding (AMR) after the term “antimicrobial resistance” is used then spell out “antimicrobial use” in L14 and add () around (AMU).

L36 – the word “likely” is not needed, may is sufficient.

Table 1 – check the population exclusion criteria? Should this be excluding non-captive animals?

L163 – should this be blinded? If so, how was it blinded?

L295 – the words “with it” are not needed, it reads awkwardly.

L378 – China have banned antibiotics as growth promotors: https://www.nature.com/articles/d41586-020-02889-y.

L411-425 – could mention the new EU regulation on group preventative use for farm animals here. In addition, Denmark have strict rules on antibiotic use for farmed animals, probably explaining the number of studies.

L489-594 – organic farming and antibiotic use. I think there could be more explanation here. Many organic farming practices (although not everywhere) include higher welfare standards, lower stocking densities, avoiding painful procedures like tail docking, and the use of enrichment. It’s not surprising that they require less antibiotics. The following report has a nice section on organic farming: “EMA and EFSA Joint Scientific Opinion on measures to reduce the need to use antimicrobial agents in animal husbandry in the European Union, and the resulting impacts on food safety (RONAFA)” (section 4.2.3.10 and appendix J).

Author Response

Reviewers’ comments are listed below, and authors’ responses are shown beneath each comment. Changes in the marked revised manuscript are highlighted in yellow.

Reviewer 1

This is an important and much needed review. It is very well constructed and written. There are some minor comments/suggestions for improvement.

Answer: We would like to thank the reviewer for the professional work in reviewing our manuscript and we are delighted to know that our work was appreciated.

General comments:

  1. Worth mentioning the new restrictions on preventative antibiotic use in the EU?

In my view, the potential for restricted treatments in organically farmed animals is worth mentioning but over-emphasised… There is a risk of withholding antibiotics in any farmed animal due to withdrawal periods for milk and meat for example.  I do agree that certain labelling, like raised without antibiotics have potential to lead to withholding of antibiotics when needed, but most organic labels don’t prohibit use, they would follow the same withdrawal periods as those used in conventional farming. This is why it’s worth mentioning the new restrictions in the EU, since all farming could end up following similarly strict AMU rules to that of organic farming to tackle the issue of AMR, and this could also lead to limited use when needed making welfare improvement very important.

Answer: We thank the reviewer for this comment. Indeed, it is true that the new veterinary prescription rules are relevant to this section. We agree with your suggestion, and we have reflected this in the revised version of the manuscript (See L436-439 and L532-538).

Line by line comments:

  1. L11-12 – AMU is there but not spelled out. To be clear, suggest adding (AMR) after the term “antimicrobial resistance” is used then spell out “antimicrobial use” in L14 and add () around (AMU).

Answer: Amended. However, we decided to not add the abbreviation (AMR) in L11 because the term ‘antimicrobial resistance’ is not used anywhere else in the simple summary, thus not requiring the use of an abbreviation.

  1. L36 – the word “likely” is not needed, may is sufficient.

Answer: Amended.

  1. Table 1 – check the population exclusion criteria? Should this be excluding non-captive animals?

Answer: We thank the reviewer for spotting this mistake. We addressed the issue by providing the right information. See Table 1 in L161.

  1. L163 – should this be blinded? If so, how was it blinded?

Answer: Yes, we confirm that the selection process was blinded as stated in L169-170 of the manuscript. Each assessor (i.e., each author) independently evaluated all the title/abstracts retrieved for this systematic review. The full list of records was available to both authors, who autonomously screened the abstracts on two separated computers.

  1. L295 – the words “with it” are not needed, it reads awkwardly.

Answer: Amended. See L 313.

  1. L378 – China have banned antibiotics as growth promotors: https://www.nature.com/articles/d41586-020-02889-y.

Answer: We thank the reviewer for this clarification. We removed China from the list of countries in L396.

  1. L411-425 – could mention the new EU regulation on group preventative use for farm animals here. In addition, Denmark have strict rules on antibiotic use for farmed animals, probably explaining the number of studies.

Answer: We agree with the reviewer. Thus, we added this information to the manuscript (See L436-439 and L444).

  1. L489-594 – organic farming and antibiotic use. I think there could be more explanation here. Many organic farming practices (although not everywhere) include higher welfare standards, lower stocking densities, avoiding painful procedures like tail docking, and the use of enrichment. It’s not surprising that they require less antibiotics. The following report has a nice section on organic farming: “EMA and EFSA Joint Scientific Opinion on measures to reduce the need to use antimicrobial agents in animal husbandry in the European Union, and the resulting impacts on food safety (RONAFA)” (section 4.2.3.10 and appendix J).

Answer: We thank the reviewer for clarifying this aspect. We improved the section by adding extra information as suggested by the reviewer. See L519-523 and L532-538.

Reviewer 2 Report

GENERAL OVERVIEW: the authors aimed to summarise, through a systematic review, the actual research with regard between animal welfare and AMU (misuse of medications). This approach is fundamental, mainly concerning the global threat of antibiotic resistance. The novelty is guaranteed in this manuscript. The review was well conducted, and the results were very evident and concise. However, some missing points regarding the PRISMA statement must be checked. I recommend accepting the manuscript; however, I would like to suggest some modifications, aiming to harmonise it with the PRISMA checklist, as follow below:

ABSTRACT: Provide a structured summary including, as applicable: background; objectives; data sources; study eligibility criteria, participants, and interventions; study appraisal and synthesis methods; results; limitations; conclusions and implications of key findings; systematic review registration number.

INTRODUCTION: In objectives, provide an explicit statement of questions being addressed concerning participants, interventions, comparisons, outcomes, and study design (PICOS).

MATERIAL AND METHODS: Describe methods used to assess the risk of bias in individual studies (including whether this was done at the study or outcome level) and how this information is used in any data synthesis. State the principal summary measures (e.g., risk ratio, mean difference). Specify any assessment of the risk of bias that may affect the cumulative evidence (e.g., publication bias, selective reporting within studies). If done, describe methods of additional analyses (e.g., sensitivity or subgroup analyses, meta-regression), indicating which were pre-specified.

RESULTS: give the results according to M&M modifications.

Author Response

Reviewers’ comments are listed below, and authors’ responses are shown beneath each comment. Changes in the marked revised manuscript are highlighted in yellow.

Reviewer 2

GENERAL OVERVIEW: the authors aimed to summarise, through a systematic review, the actual research with regard between animal welfare and AMU (misuse of medications). This approach is fundamental, mainly concerning the global threat of antibiotic resistance. The novelty is guaranteed in this manuscript. The review was well conducted, and the results were very evident and concise. However, some missing points regarding the PRISMA statement must be checked. I recommend accepting the manuscript; however, I would like to suggest some modifications, aiming to harmonise it with the PRISMA checklist, as follow below:

Answer: We would like to thank the reviewer for the professional work in reviewing our manuscript. Thank you very much for your positive and constructive feedback.

  1. ABSTRACT: Provide a structured summary including, as applicable: background; objectives; data sources; study eligibility criteria, participants, and interventions; study appraisal and synthesis methods; results; limitations; conclusions and implications of key findings; systematic review registration number.

Answer: We have revised the abstract to comply with PRISMA guidelines. The PRISMA checklist for abstracts follows in attached. However, not all requirements were applied due to space limitation (200 words) as for the Journal guidelines. See L26-40. Funding and risk of bias statements were left out of the abstract but included in other sections of the manuscript (i.e., Methods and Funding, respectively). The same is true for the synthesis method (not in the abstract but in the Methods).

  1. INTRODUCTION: In objectives, provide an explicit statement of questions being addressed concerning participants, interventions, comparisons, outcomes, and study design (PICOS).

Answer: We have updated an explicit objectives’ statement covering the PICOS, which in our case are only POs (See L128-132 of the revised manuscript).

  1. MATERIAL AND METHODS: Describe methods used to assess the risk of bias in individual studies (including whether this was done at the study or outcome level) and how this information is used in any data synthesis. State the principal summary measures (e.g., risk ratio, mean difference). Specify any assessment of the risk of bias that may affect the cumulative evidence (e.g., publication bias, selective reporting within studies). If done, describe methods of additional analyses (e.g., sensitivity or subgroup analyses, meta-regression), indicating which were pre-specified.

Answer: We further clarified the methods employed for quality assessment and for the assessment of risk bias in the Methods section (See L175-183)

  1. RESULTS: give the results according to M&M modifications.

Answer: We decided not to apply further modifications to the Results section to maintain a certain consistency with the flow of the presentation of our data. However, if the reviewer thinks that there are some specific paragraphs where some improvements can be applied, we would be happy to accept more details.
